# Acute Kidney Injury in Very Low Birth Weight Infants: A Major Morbidity and Mortality Risk Factor

**DOI:** 10.3390/children10020242

**Published:** 2023-01-29

**Authors:** Gilad Lazarovits, Noa Ofek Shlomai, Raed Kheir, Tali Bdolah Abram, Smadar Eventov Friedman, Oded Volovelsky

**Affiliations:** 1Department of Neonatology, Hadassah Medical Center, Faculty of Medicine, Hebrew University of Jerusalem, Jerusalem 9190501, Israel; 2School of Medicine, Faculty of Medicine, Hebrew University of Jerusalem, Jerusalem 9190501, Israel; 3Pediatric Nephrology Unit, Hadassah Medical Center, Faculty of Medicine, Hebrew University of Jerusalem, Jerusalem 9190501, Israel

**Keywords:** acute kidney injury, very low birth weight, patent ductus arteriosus, bloodstream infections, vasopressors, prematurity, preterm birth

## Abstract

Background and objectives: Very low birth weight (VLBW) infants are at high risk of developing acute kidney injury (AKI), presumably secondary to low kidney reserves, stressful postnatal events, and drug exposures. Our study aimed to identify the prevalence, risk factors, and outcomes associated with AKI in VLBW infants. Study design: Records of all VLBW infants admitted to two medical campuses between January 2019 and June 2020 were retrospectively reviewed. AKI was classified using the modified KDIGO definition to include only serum creatinine. Risk factors and composite outcomes were compared between infants with and without AKI. We evaluated the main predictors of AKI and death with forward stepwise regression analysis. Results: 152 VLBW infants were enrolled. 21% of them developed AKI. Based on the multivariable analysis, the most significant predictors of AKI were the use of vasopressors, patent ductus arteriosus, and bloodstream infection. AKI had a strong and independent association with neonatal mortality. Conclusions: AKI is common in VLBW infants and is a significant risk factor for mortality. Efforts to prevent AKI are necessary to prevent its harmful effects.

## 1. Introduction

Over the last few decades, there has been a significant decline in the mortality rate of preterm infants, mainly due to improved antenatal and postnatal care [1]. Consequently, the incidence of prematurity-associated complications rose dramatically. Neonatal acute kidney injury (AKI) is one of the most common hurdles in very low birth weight infants (VLBW) [2].

AKI is most commonly defined by the neonatal modified Kidney Disease: Improving Global Outcomes (KDIGO), which grades categorizes? AKI severity based on increasing creatinine levels and decreasing urine output [3,4,5]. However, only a few studies of AKI in preterm infants included urine output as part of diagnosis and severity grading. The AWAKEN study (Assessment of Worldwide Acute Kidney Injury Epidemiology in Neonates), a large multicenter study including over 2000 preterm infants, included serum creatine rise of at least 50% of the lowest value or urine output of <1 mL/kg/hour as an indicator of AKI [4]. A sub-analysis of this study has highlighted the need to adjust urine output to both gestational and chronological age [6,7]. Notably, serum creatinine level is unreliable as a glomerular filtration rate (GFR) marker in preterm infants due to its tubular reabsorption in premature kidneys and its dependence on muscle mass [8]. Cysteine C may be a superior biomarker in neonates as it does not cross the placenta, but more research is required before routine clinical use [9]. AKI is a relatively common phenomenon in neonates. In preterm infants, the risk of AKI is inversely related to gestational age. Several studies reported an incidence of 18–40% in cohorts of VLBW infants, with a higher incidence reported in infants younger than 29 weeks of gestation [10,11]. 

Several explanations exist for the relatively high rate of AKI in preterm infants. Firstly, preterm birth halts kidney development in the most vital step. Nephrogenesis ceases in utero at the 36th week of gestation due to a simultaneous and irreversible differentiation of nephron progenitor cells [12]. Approximately 60% of the nephrons in term newborns are created during the third trimester of pregnancy [13]. Although there is no nephrogenesis in term infants after birth, new nephrons are created post-birth in preterm infants up until 36 weeks of corrected gestational age [14]. However, in a study of preterm birth using a baboon model, Black et al. reported that even though there is postnatal nephrogenesis in the preterm baboon model, some of these glomeruli were abnormal, and the renal cortex was significantly thinner. They have reported similar findings in human autopsied preterm infants [15]. Ultrasonographic evaluation of renal parenchyma of preterm and term infants revealed similar findings [16]. Consequently, low nephron number in preterm newborns places them at a poor starting point regarding kidney function [17,18,19]. Secondly, newborns are more prone to AKI due to the low capacity of urine concentration due to immaturity of the tubular system and the high rate of insensible water loss due to relatively high body surface [20]. Thirdly, the glomerular filtration rate (GFR) is substantially low in the first days of life and increases gradually during the following weeks after birth [21]. Fourthly, blood flow to the neonatal kidney is relatively low and almost double in the first two postnatal weeks and continues to increase in the first years of life [22,23]. Low nephron number, poor concentration ability, inferior GFR, and blood flow, combined with treatment in neonatal intensive care units, exposure to infections, medications, and hemodynamic instability contributing to hypoperfusion and hypoxia may substantially increase the risk of AKI in this population [3]. Furthermore, term and preterm newborns are at the highest risk of developing AKI during the first postnatal days, as they are born with high renal vascular resistance, high plasma renin activity, and decreased intercortical perfusion and sodium reabsorption along the proximal tubules [24]. Therefore, preventing further damage to the kidney in the postnatal period is extremely important in this population. 

Among preterm infants, the risk of AKI is inversely correlated with gestational age (GA), with an estimated incidence as high as 48% in extremely premature infants (<29th gestational week) [10,11,25,26]. Other perinatal and postnatal factors increase AKI risk in this population even further. These factors include the exposure of preterm infants to extreme stress, such as hemodynamic instability, mechanical ventilation, bloodstream infections (BSI), necrotizing enterocolitis (NEC), and nephrotoxic drugs. A complete course of antenatal steroids decreased the incidence of AKI in VLBW preterm infants. Infants not exposed to antenatal steroids may require closer observation of their renal function [27]. A recent study of early near-infrared spectroscopy (NIRS) in the first day of life found an association between low renal early renal oxygen saturations and the development of AKI in preterm infants born before completing 32 weeks of gestation [28].

The possible short-term complications of AKI in VLBW infants include a higher risk of fluid overload [29,30], metabolic acidosis, electrolyte imbalance, drug toxicity, and inaccurate dosage adjustments. Furthermore, renal replacement therapy [31] that has been shown to reduce mortality in adults, children, and term neonates [32,33] is challenging in this population of VLBW preterm infants. Peritoneal dialysis may be used in neonates as small as 830 g [34]. However, in some cases, peritoneal dialysis is not suitable due to abdominal complications like enterocolitis and a bowel perforation. In addition, hemodialysis and continuous veno-venous hemodiafiltration are challenging due to low blood flow, low-diameter tubing, and central lines [35].

Numerous studies have shown correlations between AKI and other prematurity-related complications in VLBW infants. For instance, newborns with AKI have a higher independent risk of bronchopulmonary dysplasia (BPD) [36], neurological impairment, and interventricular hemorrhage (IVH) [3,37], and as a result, higher mortality rates. Patent ductus arteriosus (PDA) presents a significant challenge when examining the issue of AKI. Untreated PDA may lead to hemodynamic changes that may cause AKI. However, traditional treatments for PDA, such as non-steroidal anti-inflammatory drugs, may also lead to AKI in preterm infants [38]. 

In this study, we aimed to improve our understanding of the specific risk factors and mechanisms that may reduce the incidence of AKI in this population. Using data from over 150 VLBW newborns from two different campuses, we examined the association between prematurity-related factors and AKI in VLBW newborns and assessed the impact of AKI on neonatal mortality. 

## 2. Materials and Methods

### 2.1. Study Design

Electronic files of VLBW infants in two Neonatal Intensive Care Units (NICU) at Hadassah Hebrew University Medical Center in Jerusalem, Israel, were reviewed. The study included VLBW infants admitted to NICUs between January 2019 and June 2020. In both NICUs, identical medical protocols are conducted. The same medical personnel rotate between these two centers and practice a similar clinical approach. Infants were excluded if they had chromosomal anomalies, congenital kidney or urinary tract anomalies, as were infants with less than four measured creatinine values and infants who survived less than 48 h. The Institutional Review Board approved this study, and a waiver from informed consent was granted (HMO 0248-20).

### 2.2. Data Collection

Perinatal data included gestational age, birth weight, gender, and Apgar scores at 1 and 5 min. Characteristics of the hospital course included the need for vasopressor support and exposure to nephrotoxic drugs as defined in the Baby The Nephrotoxic Injury Negated by Just-in-time Action (NINJA) study [39]. (Gentamycin, Vancomycin, Nonsteroidal anti-inflammatory drugs (NSAIDs), and Piperacillin/ Tazobactam), bloodstream infections (BSI), phototherapy treatment, respiratory diseases such as respiratory distress syndrome (RDS), bronchopulmonary dysplasia (BPD), defined as the need for supplemental oxygen or positive pressure at 36 + 0 corrected age after at least 28 cumulative days [40], necrotizing enterocolitis (NEC), defined as Bell’s stage IIa or higher [41], patent ductus arteriosus (PDA), and death.

### 2.3. Acute Kidney Injury

According to the National Institute of Diabetes and Digestive and Kidney (NIDDK) neonatal AKI workshop definition [42], AKI was defined as an increase in serum creatinine of 50% from the lowest creatinine value after the first 48 h of life. When an infant had creatinine less than 50 mmol/L, AKI was not considered. Urine output was not used as a parameter due to insufficient data. Pediatric equations for estimating glomerular filtration rate were not used as they were not evaluated in the neonatal population. 

### 2.4. Outcomes

The primary outcome was the mortality rate during hospitalization and its correlation with AKI. Secondary outcomes were defined as neonatal morbidities, including BPD and length of hospitalization in the intensive care unit. Accordingly, these variables were compared to VLBW infants without AKI. 

### 2.5. Statistical Analysis

Demographic and perinatal characteristics were compared between newborns with and without AKI using a *t*-test for continuous variables and Chi-square or Fisher’s exact test for continuous and categorical variables, respectively. Multivariate logistic regression models were constructed to evaluate the significant predictors for the different variables. Variables were chosen for the model based on the significant factors from the initial univariate analysis. All comparisons utilized a two-sided significance level of 0.05. Kaplan–Meier survival estimate was sought, accompanied by the log-rank test comparing survival curves to compare the effect of AKI on death. 95% confidence interval (CI) accompanied all comparable estimates. Data analysis was performed using IBM SPSS V.26.

## 3. Results

### 3.1. Study Group

182 VLBW infants were enrolled during the 18-month study period. Thirty infants were excluded from the study. Of the remaining 152 infants, 32 (21%) developed AKI (Figure 1). Of these infants, 54% were males. The average gestational age at birth was 28.5 ± 2.9 weeks, with an average birth weight of 1073 ± 259 g. 

### 3.2. Characteristics of AKI Patients

Compared with newborns without AKI, those with AKI were more likely to be born at lower gestational age (26.4 GA vs. 29.1 GA), lower birthweight (906 g vs. 1118 g), and lower Apgar score in 1 and 5 min. However, there were no differences in cord PH and body temperature upon admission to the intensive care unit (Table 1).

The first episode of AKI occurred in the first ten days of life in the majority of cases (median nine days interquartile range [IQR] = 5–17 days).

### 3.3. Outcome

Thirteen infants died (8.5%) before NICU discharge. Mortality was significantly higher in infants with AKI than in those without AKI (31% vs. 2.5% *p* < 0.001). Eleven of thirteen infants died within the first 40 days of life. Survival curves using the Kaplan–Meier method of the two groups, with and without AKI, are presented in Figure 2. Log-rank test has also demonstrated significantly increased mortality in the AKI group. Although no significant statistical difference in the incidence of BPD was shown between the groups (*p* < 0.057), there were significantly more infants who received postnatal corticosteroid treatment for BPD in the AKI group (Table 2).

#### 3.3.1. Risk Factors Associated with AKI—Univariate Analysis

AKI was found to be significantly associated with an Apgar score of less than seven in five minutes (*p* < 0.009), PDA (*p* < 0.001), RDS (*p* < 0.012), bloodstream infection (BSI) (*p* < 0.001), hemodynamic shock that required treatment with vasopressors (*p* < 0.001), NEC (*p* < 0.001), and exposure to two or more nephrotoxic drugs (Table 1). 

#### 3.3.2. Risk Factors Associated with AKI—Multivariate Analysis

Multivariant analysis by stepwise forward test has been performed to identify the main variables that predict AKI. In this model, we tested exposure to two or more nephrotoxic drugs, RDS, PDA, use of vasopressors for hemodynamic instability, bloodstream infection (BSI), NEC, Apgar score at 5 min, GA, and BW. Based on this model, we found that hemodynamic instability, PDA, and BSI were the most significant predictors for the development of AKI (Table 3). In addition, specificity was 97.5%, and sensitivity 40.5%.

#### 3.3.3. Multivariate Analysis—Risk Factors for Mortality in VLBW

Multiple variables were associated with mortality. A multivariant analysis was performed to examine the main predictors of mortality in VLBW infants hospitalized in the NICU. Using forward stepwise analysis, we found that the two most significant factors that may predict death were hemodynamic instability that required vasopressors treatment (adjusted OR 38.4) and AKI (adjusted OR 10.6) (Table 4).

## 4. Discussion

This study evaluated AKI incidence, risk factors, and mortality rate among VLBW infants in two neonatal intensive care units. In our study population, AKI incidence was 21%. We found that AKI was an independent risk factor for mortality in this population. A multivariant analysis demonstrated that vasopressor use, PDA, and BSI were the leading risk factors for AKI. In addition, lower gestational age and birth weight were significantly associated with AKI and exposure to nephrotoxic medications. 

The incidence of AKI in our study is similar to other studies evaluating AKI in VLBW infants, according to the modified Kidney Disease Improving Global Outcomes (KDIGO) criteria. For instance, Elmas et al. reported similar risk factors for AKI, including lower gestational ages, lower Apgar scores, and inotropic support [43]. In addition, exposure to treatment with non-steroidal anti-inflammatory drugs (NSAIDs) has been identified as a risk factor for AKI in a study of 70 preterm infants born prior to 34 weeks of gestation. This effect was more significant when NSAIDs were used to treat a hemodynamically significant PDA than when used as prophylaxis [44]. 

Exposure to nephrotoxic anti-microbial drugs is almost universal for VLBW preterm infants. The often-unavoidable drugs include aminoglycosides such as gentamicin and amikacin, which are used as empirical treatment, and later vancomycin, and antifungal medications. While using these drugs in preterm infants, both serum levels of antibiotics, and serum creatinine are crucial for careful dose adjustment [45].

The overall incidence of AKI in preterm infants is difficult to estimate accurately due to the various definitions of AKI in different studies.

This challenge contributes to the low diagnosis rate, recognition, and reporting of AKI in this age group. Newer studies highlight the possible role of biomarkers such as urinary epidermal growth factor and neutrophil gelatinase-associated lipocalin for a more accurate diagnosis of AKI in preterm infants [46]. 

Nonetheless, recent studies suggest a high incidence of AKI among preterm VLBW infants treated in neonatal intensive care units, with a significant increase in morbidity and mortality compared to age- and illness-matched controls [24]. For example, in a large meta-analysis including over ten thousand premature or low birth weight infants, the prevalence of AKI was 25% (95% CI 20–30%) [47]. In the AWAKEN study, the incidence of AKI among VLBW infants was 33% [26]. In a recent study of 436 preterm infants, Khyzer et al. reported that AKI was more common in the first week of life and was negatively associated with GA. In addition, the authors found an association of AKI with critical illnesses, such as sepsis and intraventricular hemorrhage, and a positive association with mortality [48]. Additional studies evaluating risk factors for AKI found an association of AKI with lower gestational age, low APGAR score, mechanical ventilation, lower birth weight, PDA, vasoactive medications, NSAIDs, and nephrotoxic medications [47]. The United States’ database of VLBW from 2000–2017 recognized the birth weight of <1000 g and gestation < 28 weeks PMA as the main risk factors for AKI in preterm VLBW infants. The reported AKI was associated with neonatal comorbidities, such as NEC, bronchopulmonary dysplasia, and intraventricular hemorrhage. In their database of over 1,200,000 VLBW infants, after controlling for confounding factors, AKI was associated with increased mortality within this population [49].

In our study, PDA, BSI, and the need for vasoactive medications were found to be predictive of AKI, with a specificity of 97.5%. This finding may imply that many cases of AKI may be preventable with improved postnatal care [50]. The most appropriate treatment for PDA in VLBW infants is currently controversial among neonatologists worldwide [51]. Majed et al. reported that moderate to large PDA was associated with AKI in preterm infants born before 28 weeks of gestation and that treatment with NSAID decreased the risk of mild but not severe AKI [52]. However, a study of 150 preterm infants born prior to 28 weeks’ gestation reported similar rates of AKI between infants with no PDA and infants with a hemodynamically significant or insignificant PDA [38].

Evidence regarding the effect of NSAID exposure in preterm infants on AKI is also conflicting. In our cohort, BSI was associated with an increased rate of AKI. This finding was reported in additional studies of AKI in neonates. In a cohort-matched study, Coggins et al. reported increased rates of AKI and greater AKI severity in neonates with culture-proven sepsis [53]. The mechanism in which sepsis contributes to the development of AKI may include decreased GFR, renal vasoconstriction, and reduced local blood flow [54].

Regarding the prevention of BSI in VLBW infants, quality improvement studies have demonstrated that reducing the incidence of bloodstream infections is a feasible goal [55,56].

Both PDA and BSI are factors that contribute to hemodynamic instability in VLBW infants. A single-center study of 59 term and preterm infants reported a different clinical profile of AKI in term and preterm infants. The latter was characterized by later onset and milder but recurrent episodes [57]. 

The primary outcome of this study was to explore the association between AKI and mortality. We found that VLBW infants with AKI had significantly higher mortality rates, with an adjusted odds ratio of 10.6 (95% CI 1.98–56.99). According to our study, the prevalence of death among VLBW with AKI was 12.4 times higher than in the control group. AKI was the second most significant risk factor, following vasopressor use. Additional studies report similar findings of higher mortality rates among VLBW infants with AKI [26,47].

The mechanism of high mortality in neonatal AKI remains obscure. Renal failure has multisystemic effects due to cross-talk with other organs, such as the heart, liver, immune system, and lungs. AKI-associated fluid overload and acid–base imbalance can attenuate cardiac, pulmonary, or liver function and contribute to general deterioration. In addition, patients with AKI have massive inflammatory mediator activation [58]. Cardio-renal syndrome has also been proposed in the pediatric population [59]. 

Arcinue et al., in a study of extremely low birth weight infants over ten years, found an AKI prevalence of 26%, with a twofold mortality rate compared to infants without AKI within the same cohort [60]. An additional study including 229 VLBW infants supports the observation of higher mortality rates of VLBW infants with AKI [61]. A recent study of preterm infants born prior to 28 weeks’ gestation reported higher mortality and longer length of stay among infants with AKI, more so with recurrent episodes of AKI [62]. Askenazi et al. reported a significant increase in mortality and bronchopulmonary dysplasia in preterm infants who developed AKI during their NICU stay compared to those who did not [63]. A recent multi-center study of 900 extremely low gestation infants found AKI in 19% of subjects. Stage three AKI was associated with a 4-fold incidence of death, with severe stage two AKI associated with a 2.2-fold hazard of death [2].

There is growing evidence that AKI in preterm infants who survive the NICU course has significant implications on renal health in adolescence and adulthood [64,65]. This finding highlights the importance of early recognition, treatment, and long-term follow-up of AKI in preterm infants. The rate of progression from AKI to chronic kidney disease (CKD) later in life is poorly known among the VLBW infant population. It is reported that children born prematurely have a 3-fold increased risk of chronic kidney disease and a 1.5-fold higher risk of end-stage kidney disease over their life course than children born full term [66]. Moreover, these infants are at increased risk of developing systemic hypertension, type two diabetes, metabolic syndrome, heart failure, and ischemic heart disease in adulthood, all of which have a direct impact on renal function [67]. In addition, it is important to address the high potential for glomerular sclerosis, given the challenges of maladaptive renal repair, ongoing exposure to inflammation, abnormal regeneration, and reduced nephron number [68]. The issue of CKD will continue to grow as more VLBW infants survive into adulthood. In this view, extensive multicenter long-term follow-up studies of VLBW infants that developed AKI are required to understand the potential risk for CKD better. 

Interventions to avoid postnatal nephron loss and improve the long-term renal health of preterm infants are explored. These include avoiding neonatal AKI and nephrotoxin exposure after birth, educating preterm survivors and their families about lifestyle risks that may contribute to nephron loss, and educating healthcare providers about the importance of reducing obesity, screening for hypertension and insulin resistance, and avoiding nephrotoxins in previously born preterm children and adults. The evidence on optimal nutrition for nephrogenesis is limited; however, human milk feeding has been associated with a reduction in hypertension in adolescents born preterm in nonrandomized studies [69]. A potential treatment opportunity relies on the presence of many active endogenous stem cells in the preterm kidney. As pluripotent cells can generate new nephrons, regenerative postnatal and perinatal medicine may be available. These treatments may improve kidney structure and function in adolescence and adulthood [70].

The risk of AKI in preterm infants and, potentially later in life, the development of CKD suggests that preterm birth significantly impacts the “developmental programming” direction. Therefore, abnormalities in organogenesis, secondary to preterm birth, may influence structure and function throughout life.

Our study is limited by its retrospective nature, as medical records may be incomplete. Additionally, we did not use urine output to define AKI due to a lack of data. However, AKI prevalence in our study is similar to those reported in the literature using urine output and renal function tests [25]. It is noteworthy that our study included a high number of VLBW infants from two NICUs in Jerusalem. The study population was highly diverse and representative of our national population. 

Further studies are needed to determine the molecular and physiological links between AKI and death. Better and more sensitive biomarkers of AKI in preterm VLBW infants need to be researched, and specific gestational age and birth weight standards need to be established. In addition, additional quality improvement studies are required to decrease the incidence of AKI and its adverse impact on neonatal and adulthood outcomes.

In conclusion, our study demonstrated that AKI is common in VLBW infants and is significantly and independently related to neonatal mortality. Furthermore, hemodynamic instability was a major contributor to AKI in VLBW infants. Accordingly, renal function follow-up is essential in septic or hemodynamically unstable VLBW infants. In light of the evidence of lifelong sequala of AKI of prematurity, and given the high incidence, there is room to consider routine post-discharge nephrological follow-up for VLBW preterm infants.

## Figures and Tables

**Figure 1 children-10-00242-f001:**
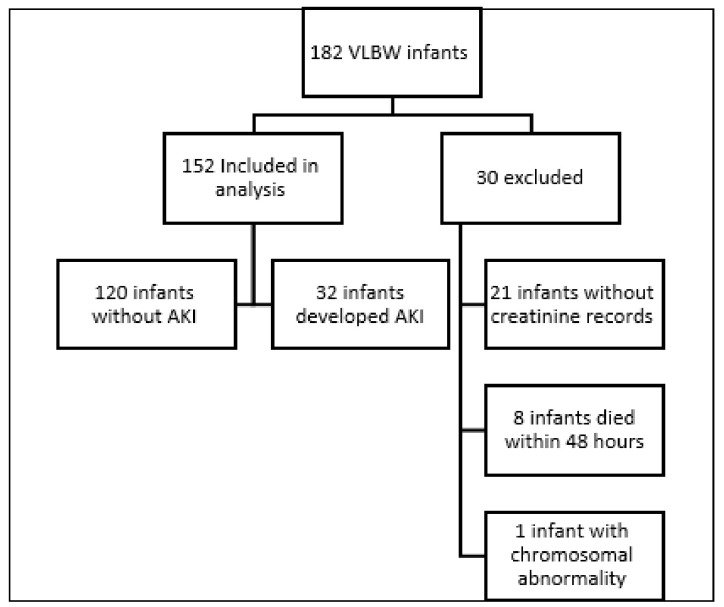
Study flow chart. VLBW—Very low birth weight. AKI—Acute kidney injury.

**Figure 2 children-10-00242-f002:**
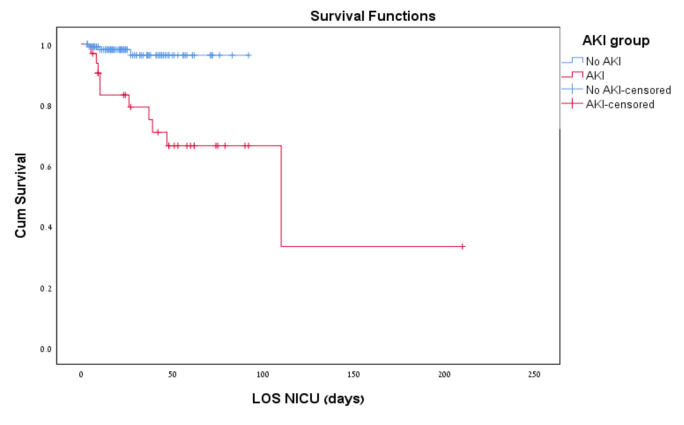
Kaplan–Meier survival curve for infants with and without AKI in the neonatal intensive care unit. AKI = acute kidney injury; LOS = length of stay.

**Table 1 children-10-00242-t001:** Patient information among VLBW infants with and without AKI, comorbidities, interventions.

	No AKI (*n* = 120)	AKI (*n* = 32)	*p*-Value
Male gender	68 (57%)	15 (47%)	0.323
GA (mean week ± SD)	29.12 ± 2.6	26.4 ± 2.7	*p* < 0.001
Birth weight (mean ± SD)	1118 ± 235	906 ± 282	*p* < 0.001
Cord PH	7.29 ± 0.08	7.32 ± 0.08	0.095
Apgar score in 1st minute <7	50 (42%)	16 (50%)	0.43
Apgar score in 5 min <7	9 (7.5%)	8 (25%)	0.01
Temperature at 15 min from admission	36.7 ± 0.53	36.3 ± 1.24	0.13
RDS	51 (42%)	21 (65%)	0.012
TTN	15 (12.5%)	1 (3.2%)	0.308
PDA	21 (17.5%)	17 (53%)	*p* < 0.001
Ibuprofen for PDA closure	3 (14% of infants with PDA)	2 (11% of infants with PDA)	1
Vasopressors medications	6 (5%)	12 (37.5%)	*p* < 0.001
BSI	5 (4%)	9 (28%)	*p* < 0.001
NEC	12 (10%)	13 (41%)	*p* < 0.001
Surgical NEC	1 (1%)	9 (28%)	0.01

VLBW = very low birthweight; AKI = acute kidney injury; GA = gestational age; SD = standard deviation; RDS = respiratory distress syndrome; TTN = transitory tachypnea of the newborn; PDA = patent ductus arteriosus; BSI = bloodstream infection; NEC = necrotizing enterocolitis.

**Table 2 children-10-00242-t002:** Patient outcomes.

	No AKI (*n* = 120)	AKI (*n* = 32)	*p*-Value
BPD	34 (28%)	15 (47%)	0.057
Postnatal Steroids for BPD	20 (17%)	11 (34%)	0.035
Death	3 (2.5%)	10 (31%)	*p* < 0.001

AKI = acute kidney injury; BPD = bronchopulmonary dysplasia.

**Table 3 children-10-00242-t003:** Multivariate analysis by stepwise forward test for prediction of acute kidney injury in very low birth weight infants.

	Significance	Adjusted OR	95% Confidence Interval
Vasopressors	0.003	6.81	(1.88, 24.6)
PDA	0.012	3.62	(1.32, 9.93)
BSI	0.025	4.77	(1.21, 18.78)

PDA = patent ductus arteriosus; BSI = bloodstream infection.

**Table 4 children-10-00242-t004:** Multivariate analysis by stepwise forward test for predicting mortality in very low birth weight infants.

	Significance	Adjusted OR	95% Confidence Interval
Vasopressors	0.000	38.48	(7.26, 204)
Acute kidney injury	0.006	10.6	(1.98, 56.99)

## Data Availability

Data is unavailable due to privacy.

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
