# Peer review of "Acute Kidney Injury in Very Low Birth Weight Infants: A Major Morbidity and Mortality Risk Factor"

_children, 2023, doi:10.3390/children10020242_

Round 1

Reviewer 1 Report

In this manuscript, the authors investigate the prevalence of acute kidney injury in very low birth weight neonates and the impact of AKI on mortality. The authors looked at risk factors that might contribute to AKI in this population.

While the study design is appropriate, it does look at single-center data with its attendant drawbacks.

This reviewer has one major concern: The method of estimating renal function using serum creatinine is not really optimal. The standard is to use either the original Schwartz equation or the Counahan-Barratt equation. Additionally, there should be a correction for preterm neonates. This reviewer would like this addressed.

Author Response

Please read the attached document.

Reviewer 2 Report

In this work, the authors identify the prevalence, risk factors, and outcomes associated with AKI in VLBW infants. Several suggestions are made as follows to improve the quality of the manuscript.
1. All abbreviations should be substantiated for the first time.
2. The manuscript needs thorough editorial corrections with special attention to linguistic improvement, punctuation (line 79), bold font and table format (table1,2) as well as reference format (double number in each reference)

3. What is the number of cases in the two hospitals in your study?Have you compared the results of the two hospitals? how do you think the situation in the two hospitals can represent the situation in Israel?

4.The mortality and risk factors in this study have been evaluated in neonates, preterm infants and very low birth weight infants who develop AKI. What are the highlights of this study?

Author Response

Please read the attached document. 
